# Quantitative analysis of 1300-nm three-photon calcium imaging in the mouse brain

Tianyu Wang[1][†]*, Chunyan Wu[1,2][†], Dimitre G Ouzounov[1], Wenchao Gu[3], Fei Xia[4], Minsu Kim[5], Xusan Yang[1], Melissa R Warden[3], Chris Xu[1]*

[1]School of Applied and Engineering Physics, Cornell University, Ithaca, United States; [2]College of Veterinary Medicine, Cornell University, Ithaca, United States; [3]Department of Neurobiology and Behavior, Cornell University, Ithaca, United States; [4]Meining School of Biomedical Engineering, Cornell University, Ithaca, United States; [5]College of Human Ecology, Cornell University, Ithaca, United States

**Abstract** 1300 nm three-photon calcium imaging has emerged as a useful technique to allow calcium imaging in deep brain regions. Application to large-scale neural activity imaging entails a careful balance between recording fidelity and perturbation to the sample. We calculated and experimentally verified the excitation pulse energy to achieve *the minimum photon count* required for the detection of calcium transients in GCaMP6s-expressing neurons for 920 nm two-photon and 1320 nm three-photon excitation. By considering the combined effects of in-focus signal attenuation and out-of-focus background generation, we quantified the cross-over depth beyond which three-photon microscopy outpeforms two-photon microscopy in recording fidelity. Brain tissue heating by continuous three-photon imaging was simulated with Monte Carlo method and experimentally validated with immunohistochemistry. Increased immunoreactivity was observed with 150 mW excitation power at 1 and 1.2 mm imaging depths. Our analysis presents a translatable model for the optimization of three-photon calcium imaging based on experimentally tractable parameters.

*For correspondence:
tw329@cornell.edu (TW);
cx10@cornell.edu (CX)

[†]These authors contributed equally to this work

Competing interests: The authors declare that no competing interests exist.

## Introduction

Multiphoton microscopy combined with genetically encoded calcium indicators (GECIs) is a powerful functional imaging technique widely applied to in vivo neurophysiological recordings (*Lin and Schnitzer, 2016*; *Yang and Yuste, 2017*). In the mouse brain, 2-photon microscopy (2PM) has enabled activity recording from thousands of neurons with single-cell resolution (*Sofroniew et al., 2016*; *Stirman et al., 2016*; *Weisenburger et al., 2019*). Although 2-photon excitation (2PE) can effectively reduce out-of-focus fluorescence, the background intensity eventually becomes comparable to the signal in non-sparsely labeled samples, as the excitation power grows exponentially with imaging depth. The background not only reduces image contrast but also introduces additional noise inseparable from the true calcium transient signal, which irreversibly reduces imaging quality. In the mouse neocortex, 2PM signal-to-background ration (SBR) decreases to one at ~4.7 attenuation lengths with a labeling density of ~2% (*Kobat et al., 2011*; *Theer et al., 2003*; *Theer and Denk, 2006*).

3-photon microscopy (3PM) has emerged as a useful tool for imaging in deep brain regions that are typically inaccessible to 2PM. Although deep calcium imaging with 2PM has been demonstrated with red-shifted calcium indicators using 1100 nm excitation, these methods suffer from limitations such as staining only a single layer of the cortical neurons (*Tischbirek et al., 2015*), or in the hippocampus of young mice less than 6 weeks old (*Inoue et al., 2019*; *Kondo et al., 2017*). Furthermore,

the best red-shifted GECIs are currently less bright and robust than their green counterparts (e.g., <40% the cross section of GCaMP6s and <50% sensitivity of jGCaMP7; *Dana et al., 2019*; *Dana et al., 2016*), which further limits their application to large-scale recordings. Compared to the red GECIs, the shorter 2PE wavelength (900–960 nm) for GCaMP results in more tissue scattering (*Jacques, 2013*). Consequently, 2-photon imaging with GCaMP faces more technical challenges in the deep brain, and usually requires the removal of the superficial cortex or the insertion of penetrating optical elements (*Attardo et al., 2015*; *Dombeck et al., 2010*; *Low et al., 2014*; *Pilz et al., 2016*). In comparison, 1300 nm 3PM has enabled activity imaging with GCaMP6 of the entire densely labeled cortex (*Takasaki et al., 2020*; *Yildirim et al., 2019*) and the hippocampus in intact *adult* mouse brains (8–16 weeks old) (*Ouzounov et al., 2017*; *Takasaki et al., 2020*; *Weisenburger et al., 2019*) because of the background suppression by 3-photon excitation (3PE) and the reduced tissue attenuation by the longer excitation wavelength (*Ouzounov et al., 2017*).

Since the first demonstrations of 3PM for in vivo brain imaging (*Horton et al., 2013*; *Ouzounov et al., 2017*), a number of research groups have successfully adopted and developed the technology (*Bi et al., 2018*; *Escobet-Montalbán et al., 2018*; *Perillo et al., 2017*; *Rodríguez et al., 2018*; *Rowlands et al., 2017*; *Takasaki et al., 2020*; *Weisenburger et al., 2019*; *Yildirim et al., 2019*), propelled by the commercially available lasers and microscopes. To facilitate 3PM applications, here we present the quantitative characterization results of 3-photon deep-brain calcium imaging, with a side-by-side comparison to 2PM with 920 nm excitation. Our results show that 1320 nm 3PM is advantageous in terms of both signal strength and imaging contrast in the deep cortex and beyond. Furthermore, by delineating the power constraints, we provide a framework to optimize 3-photon imaging parameters to push the limit of 3PM penetration depth and field-of-view (FOV).

## Results

### Signal photon count provides the fundamental metric for calcium imaging quality

High-quality calcium imaging enables reliable detection of calcium transients in the presence of noise. With a photomultiplier tube as the fluorescence detector, calcium imaging can be performed at the photon shot noise limit, where the photon fluctuation noise ($N$) and the signal ($S$) obey the relation: $N = \sqrt{S}$. Therefore, quantifying the fluorescence signal with photon counts provides the noise statistics, and allows probabilistic inference of the true underlying signal. Assuming exponential decay of calcium transients, a single discriminability index ($d'$), as defined by *Equation 1*, has been derived in the past to assess the recording fidelity of a calcium transient (*Wilt et al., 2013*):

$$d' \approx \frac{\Delta F}{F} \sqrt{\frac{F_0 \tau_{1/e}}{2}} \tag{1}$$

where $\tau_{1/e}$ is the 1/e decay time, $\Delta F/F$ is the peak fluorescence change of a single-action-potential-induced calcium spike, and $F_0$ is the baseline brightness of the neuron. This expression holds as long as the frame rate is high enough to sample the exponential decay of the fluorescence intensity, and a higher $d'$ value indicates better calcium transient detection accuracy. In fact, the fidelity of calcium imaging is determined by the signal photon counts of each neuron. The *minimum photon counts* required to detect a calcium transient induced by a single action potential can be calculated at any given confidence level: using the parameters for GCaMP6s ($\Delta F/F \sim 30\%$ and $\tau_{1/e} \sim 2$ s) (*Chen et al., 2013*), $F_0$ is calculated to be ~100 photons/second to achieve $d' = 3$, which allows 93% true detection and 7% false-positive detection rate. For large calcium transients induced by a short burst of multiple action potentials, $F_0$ can be significantly lower since $\Delta F/F$ is larger by the accumulation of single-action-potential induced $\Delta F/F$ (*Wilt et al., 2013*). Given $\Delta F/F$ and $\tau_{1/e}$ of a calcium indicator are fixed, the quality of calcium imaging can only be improved by increasing the baseline neuron brightness $F_0$, which can be achieved by adjusting a number of imaging parameters, for example excitation repetition rate, pulse duration, pulse energy, focal spot size and laser dwell time on each neuron (a more detailed analysis is presented in Discussion). It is clear from the above analysis that extremely high sampling rates in either space (e.g., the number of pixels) or time (e.g., the frame

rate) do not improve the overall calcium transient detection accuracy since the number of photons per neuron per second is conserved regardless of the sampling scheme.

## 1320 nm 3PE is more power-efficient in signal generation than 920 nm 2PE in the deep cortex and beyond

In general, due to the higher-order nonlinear excitation, 3PE requires higher excitation intensity at the focus in order to generate the same amount of fluorescence as 2PE. For deep tissue imaging, however, the pulse energy at the brain surface can be less for 3PE than 2PE because photons at the longer wavelength used for 3PE experience less attenuation. To quantify the difference in tissue attenuation caused by wavelength, we imaged mouse brain vasculature uniformly labeled with fluorescein dextran and measured fluorescence signal decay as a function of depth. We centered the 3PE and 2PE excitation spectra at 1320 nm and 920 nm, respectively, since they are nearly optimal for GCaMP6 imaging (*Ouzounov et al., 2019*). Through simultaneous 2PM and 3PM imaging of the same blood vessel with the same fluorescent label, we ensured the same signal collection efficiency for the two imaging modalities, and any sample fluctuation over time is eliminated. As shown in *Figure 1A*, the effective attenuation length (EAL) at 1320 nm in the mouse cortex is almost twice of that at 920 nm (297 ± 11 µm vs. 153 ± 10 µm, mean ± standard deviation, n = 4, >12 weeks old mice, all males, *Figure 1A* and *Figure 1—figure supplement 1*).

We measured the pulse energy required for detecting 0.1 photon per excitation pulse, which is a typical signal strength for multiphoton imaging (*Figure 1B*). To generate the same amount of signal in fluorescein-labeled blood vessels located at the brain surface, 3PE requires about 15 times the pulse energy of 2PE (1.1 ± 0.03 nJ vs. 0.07 ± 0.004 nJ, mean ± 95% confidence interval). As the imaging depth increases, to maintain the same signal strength, the 2PE pulse energy delivered to the brain surface has to increase more rapidly than that for 3PE since 920 nm photons experience more tissue attenuation than 1320 nm photons. Eventually, 3PE becomes more power-efficient than 2PE at a depth of around 750 µm, which is defined here as the cross-over depth. The same trend can be observed in GCaMP6s-labeled neurons. At the brain surface, 3PE at 1320 nm requires ~8 times the pulse energy as 2PE at 920 nm (1.86 ± 0.27 nJ vs. 0.24 ± 0.05 nJ, mean ± 95% confidence interval), while in cortical layers at the depth around 700 µm, the pulse energy required for 3PE becomes comparable to that of 2PE at the brain surface. The measured ratio of the 3PE to 2PE pulse energy and the cross-over depth are in good agreement with the calculation based on the typical value of 2PE and 3PE cross sections (Appendix 1). Our data show that, with the same repetition rate and pulse duration, 3PM is more power-efficient than 2PM when imaging in the deep cortex and beyond. In other words, even if 2PM were used to image at the same depth (e.g.,~1 mm), its repetition rate will have to be reduced to the same level as 3PM (typically ~1 MHz) to avoid tissue heating by the average power. This result is consistent with previous studies on deep brain 2-photon imaging, where the repetition rates were reduced to below 1 MHz at ~1 mm imaging depth (*Theer et al., 2003*; *Wang et al., 2018a*). Although our data were measured with fluorescein and GCaMP6s, the conclusion is likely applicable to other green fluorescent dyes.

## 1320 nm 3PM has orders of magnitude higher SBR than 920 nm 2PM for deep imaging in the non-sparsely labeled mouse brain

The imaging depth of 2PM is limited by the out-of-focus background in non-sparsely labeled samples (*Theer and Denk, 2006*). The SBR of an image is quantified as follows: the background is measured as the fluorescence intensity in unlabeled structures, and the signal is the fluorescence signal within the labeled structures minus the background. For both 2PM and 3PM, the background is negligible at the sample surface, with SBR ~100 in fluorescein-labeled blood vessels (*Figure 2A* and *Figure 2B*). As the imaging depth increases beyond 600 µm in the cortex, 920 nm 2PM background starts to increase rapidly, accompanied by apparent degradation of image quality (*Figure 2A* and *Figure 2C* lower panel). Since the SBR of 2PM depends on the staining density of the sample, we quantified the fractional vascular volume (2 ± 1%, mean ± standard deviation) and staining inhomogeneity (~50) of the labeled blood vessels (*Figure 2—figure supplement 1* – Materials and methods). According to the theoretical calculation, the SBR of 2PM reaches one at ~4.7 EALs at such a staining inhomogeneity (*Theer and Denk, 2006*), which is in close agreement with our in vivo measurement (*Figure 2B*). In contrast, 1320 nm 3PM does not show any increase in the background until

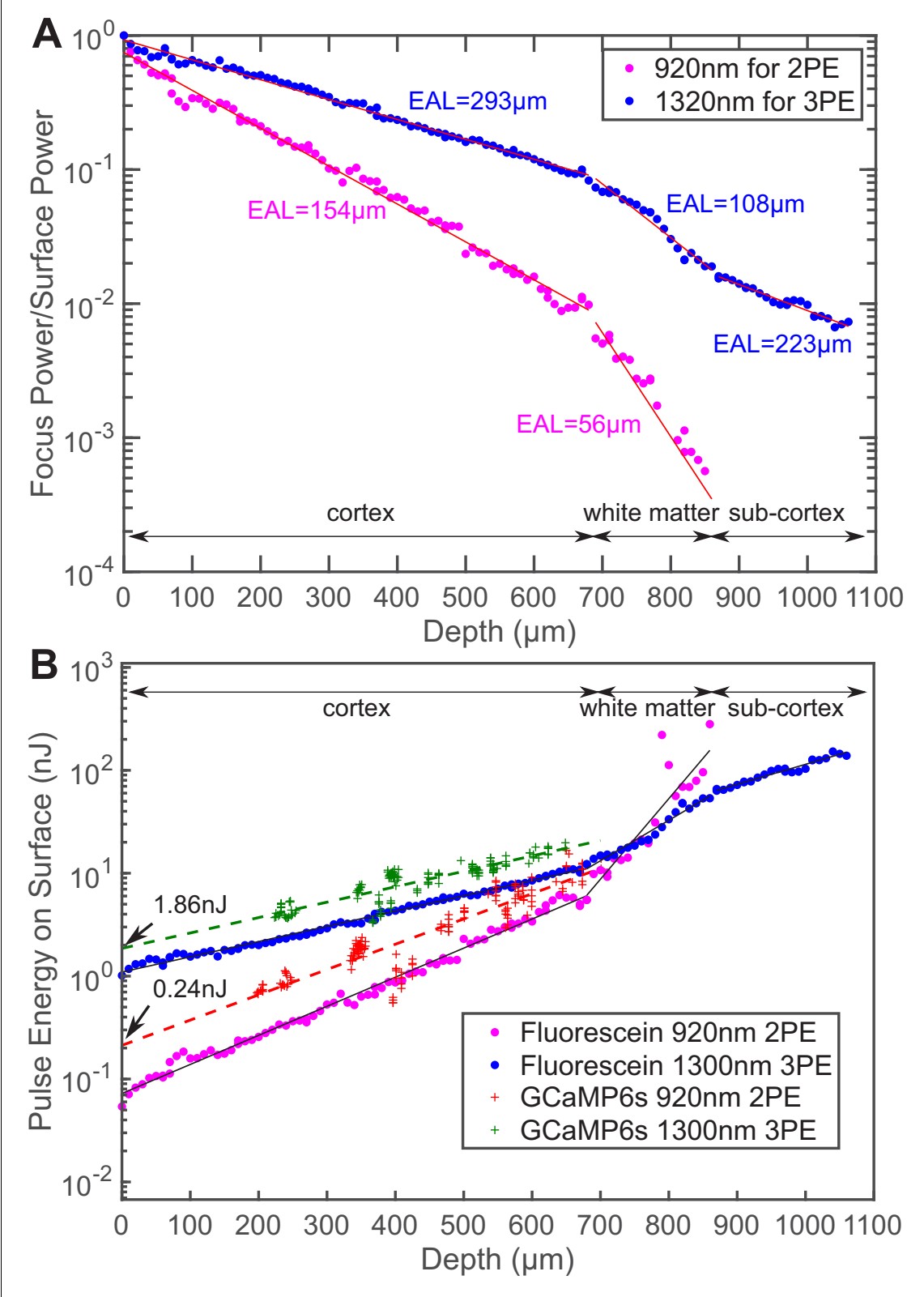

**Figure 1.** Comparison of the power attenuation of 1320 nm and 920 nm excitation light and their respective 3-photon and 2-photon excitation efficiency in the mouse brain. (**A**) Power attenuation of 920 nm and 1320 nm excitation light in the mouse brain. The mouse brain vasculature was *uniformly* labeled with fluorescein dextran and imaged *simultaneously* by 920 nm 2PM and 1320 nm 3PM at precisely the same location. The fraction of excitation power reaching the focus from the brain surface (Focus Power/Surface Power) was calculated based on the attenuation of 2PE and 3PE signal

Figure 1 continued

with the imaging depth (Materials and methods). (**B**) The pulse energy required at the brain surface to generate the *same* 2PE and 3PE signal strength (0.1 signal photon detected per laser pulse) at different imaging depths, measured in fluorescein-labeled blood vessels (n = 1 shown) and GCaMP6s-labeled neurons (CamKII-tTA/tetO-GCaMP6s; n = 5 mice). The signal strength is scaled to 60 fs pulse duration for both 2PE and 3PE.

The online version of this article includes the following source data and figure supplement(s) for figure 1:

**Source data 1.** A typical power attenuation with depth curve of 1320 nm and 920 nm excitation light in the mouse brain, plotted in *Figure 1A*.

**Source data 2.** The excitation pulse energy at the brain surface required to generate 0.1 signal photon per pulse at different depths in the mouse brain, measured in fluorescein-labeled vasculature and plotted in *Figure 1B*.

**Source data 3.** The excitation pulse energy at the brain surface required to generate 0.1 signal photon per pulse at different depths in the mouse brain, measured in the neurons of transgenic animals (CamKII-tTA/tetO-GCaMP6s) and plotted in *Figure 1B*.

**Figure supplement 1.** Additional power attenuation curves of 920 nm and 1320 nm excitation light in the mouse neocortex (n = 3, >12 weeks old mice, all males).

**Figure supplement 1—source data 1.** Additional power attenuation with depth curves of 1320 nm and 920 nm excitation light in the mouse neocortex, plotted in *Figure 1—figure supplement 1*.

in the white matter (*Figure 2A* and *Figure 2B*). The background reduction by 3PE was contributed by the longer EAL of the long wavelength that reduces the normalized imaging depth ($z$/EAL), and the higher order of nonlinearity of 3PE that suppresses background generation in the out-of-focus volume, which plays a critical role when the 2PE and 3PE use the same wavelength and have the same EAL (*Wang et al., 2018b*). Large SBRs (>40) were reported on 3-photon imaging of mouse brain vasculature at an imaging depth of greater than 5 EALs (*Liu et al., 2019*). However, for the depths beyond the white matter, our measured SBR of 3PM is lower than the theoretical prediction, which may be caused by the deterioration of the point spread function due to the strong aberration induced by the white matter. Similar behavior was also observed in through-skull imaging (*Wang et al., 2018b*).

## 3PM has higher calcium imaging sensitivity and discriminability than 2PM in the deep brain

The performance of 1320 nm 3PM calcium imaging was compared to 920 nm 2PM using a time-division multiplexing scheme, which allows essentially simultaneous recording of calcium dynamics from the same region of interest (ROI) with pixel-wise multiplexed 2PE and 3PE signals (*Ouzounov et al., 2017*) (Materials and methods). From the neurons in the shallow mouse cortical layer 2/3 and 4, we obtained nearly identical calcium traces with 2PM and 3PM (*Figure 2C* upper panel), with a Pearson's correlation factor of 0.98 ± 0.01 (mean ± standard error, calculated from 60 traces, each 75 s from neurons located from 200 to 400 μm in depth). The relative $\Delta F/F$ of GCaMP6s measured by 1320 nm 3PM is close to that of 920 nm 2PM, with the ratio $(\Delta F/F)_{2P}/(\Delta F/F)_{3P} = 1.0 \pm 0.25$ (mean ± standard deviation, n = 904 calcium transients from cells located from 240 to 450 μm), in close agreement to previous studies (*Ouzounov et al., 2019*; *Ouzounov et al., 2017*; *Takasaki et al., 2020*).

As the imaging depth increases, however, the 2PM background contributes to the increase of baseline fluorescence of neurons as $F_0' = F_0(1 + 1/\text{SBR})$, which reduces the apparent $\Delta F/F$ by $(1 + 1/\text{SBR})^{-1}$ (assuming time-invariant background). A typical comparison of 3PM and 2PM calcium traces in the deep cortex is shown in *Figure 2C* lower panel, and the ratios of 2PM to 3PM $\Delta F/F$ are shown in *Figure 2D*. In the transgenic mouse brain (CamKII-tTA/tetO-GCaMP6s) at ~4 EALs at 920 nm, the apparent $\Delta F/F$ measured by 2PM is approximately half of that by 3PM, indicating SBR ~1, which corroborates with the direct SBR measurement in *Figure 2B*. The 2P $\Delta F/F$ deteriorates rapidly with even larger imaging depth, due to the rapid decline of SBR (e.g., *Figure 2B*). Similar results have been observed in other densely labeled transgenic lines (Slc17a7/ai162) by independent studies (*Takasaki et al., 2020*). The cell-to-cell variation of the ratio of 2PM to 3PM $\Delta F/F$ is mainly caused by the difference in GCaMP expression level (i.e., dimmer cells have lower SBR than brighter cells at the same depth). Therefore, the decreasing SBR is particularly detrimental for measuring the

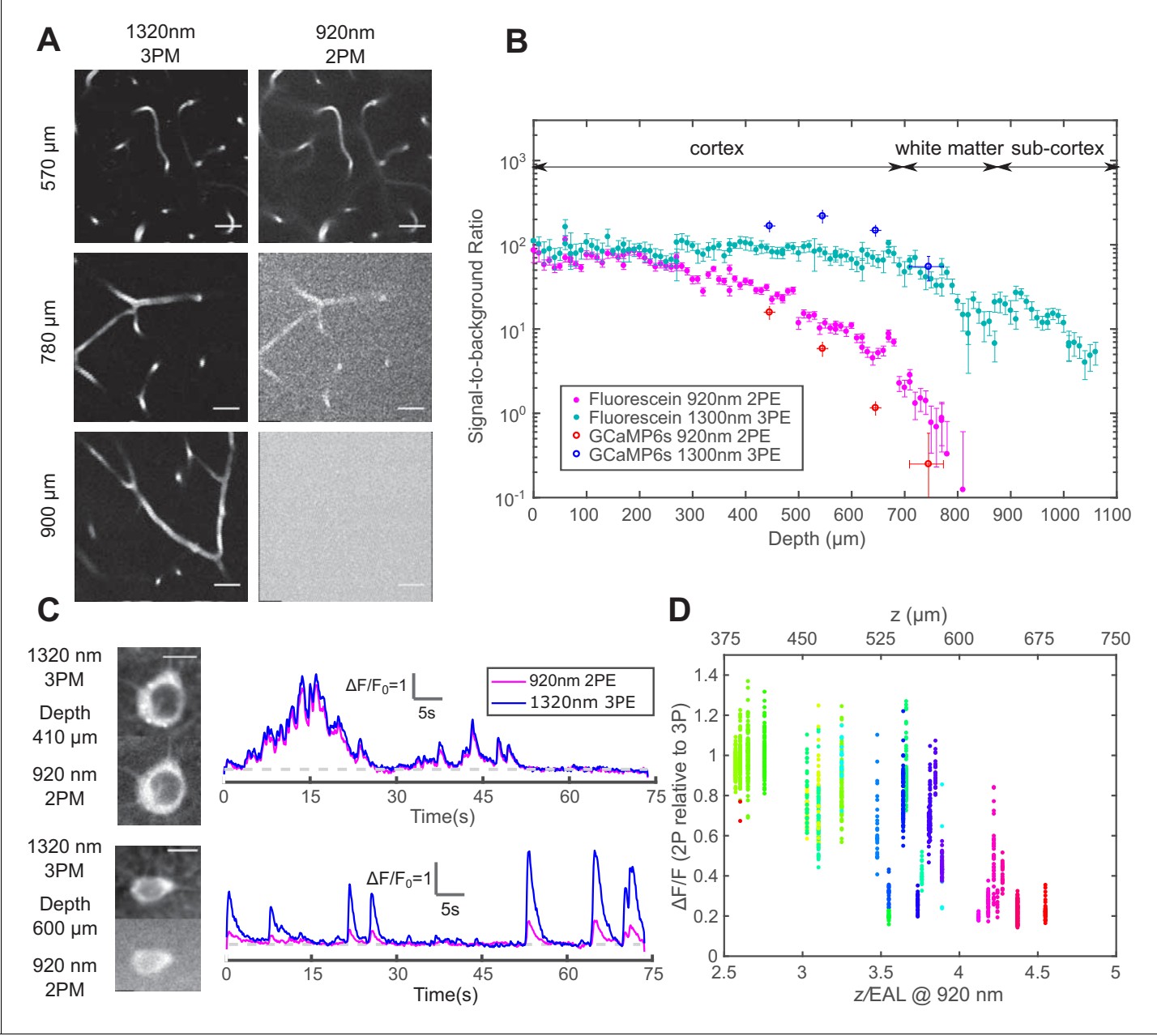

**Figure 2.** Comparison of signal-to-background ratio for 1320 nm 3PM and 920 nm 2PM in the non-sparsely labeled mouse brain and its effect on calcium imaging sensitivity. (**A**) Comparison of 3PM and 2PM images of fluorescein-labeled blood vessels at different depths. Scale bar 30 µm. (**B**) SBR measured *simultaneously* by 1320 nm 3PE and 920 nm 2PE on fluorescein-labeled blood vessels and GCaMP6s-labeled neurons. Each set of 3PE and 2PE comparison was performed in the same mouse. All vertical error bars denote the standard deviation of SBR caused by feature brightness variation, and horizontal error bars represent the depth range of the measurement. (**C**) Images and corresponding calcium traces of a cortical L4 neuron (410 µm) and a L5 neuron (600 µm), simultaneously recorded by 920 nm 2PM and 1320 nm 3PM. Both traces were low-pass filtered with a hamming window of a time constant 0.37 s. Scale bar, 10 µm. (**D**) The ratio of ΔF/F calculated on simultaneously recorded 3PM and 2PM calcium traces (CamKII-tTA/tetO-GCaMP6s; n = 3 mice). Each point was measured on a single calcium transient. The depth was normalized by the 2PM attenuation length at 920 nm of each animal to collapse the data better onto a single trendline. Data points from different neurons at the same depth are colored differently.

The online version of this article includes the following source data and figure supplement(s) for figure 2:

**Source data 1.** The change of signal-to-background ratio with depth of 1320 nm 3PM and 920 nm 2PM in the mouse brain, measured in fluorescein-labeled vasculature and plotted in *Figure 2B*.

**Source data 2.** The change of signal-to-background ratio with depth of 1320 nm 3PM and 920 nm 2PM in the mouse brain, measured in the neurons of transgenic animals (CamKII-tTA/tetO-GCaMP6s) and plotted in *Figure 2B*.

*Figure 2 continued on next page*

*Figure 2 continued*

**Source data 3.** Calcium traces recorded by 920nm 2PM on GCaMP6s-labeled neurons at different depths in transgenic animals (CamKII-tTA/tetO-GCaMP6s), based on which *Figure 2—source data 5* is derived.

**Source data 4.** Calcium traces recorded by 1320nm 3PM simultaneously on the same GCaMP6s-labeled neurons as in *Figure 2—source data 3* in transgenic animals (CamKII-tTA/tetO-GCaMP6s), based on which *Figure 2—source data 5* is derived.

**Source data 5.** The ratio of calcium transient ΔF/F between simultaneously recorded by 1320 nm 3PM and 920 nm 2PM calcium traces, on the same GCaMP6s-labeled neurons as described in *Figure 2—source datas 3* and *4*.

**Figure supplement 1.** Measurement of the staining density in uniformly labeled vasculature.

**Figure supplement 1—source data 1.** The area fraction of vasculature measured in the mouse brain, plotted in *Figure 2—figure supplement 1*.

**Figure supplement 2.** The pulse energy required at the brain surface to generate *the same d′* per pulse sampling the neuron for 2PE and 3PE of GCaMP6s at different imaging depths.

neuronal activity of the dimmer neurons for deep brain 2PM. Even at 3 EALs (~450 μm), some dimmer neurons already show lower $\Delta F/F$ for 2PM than that for 3PM (*Figure 2D*).

In addition to the loss of contrast, the noise accompanied by the 2PM background reduces the calcium transient detection accuracy, which is quantified by modifying the expression of $d'$ to include the fluorescence background:

$$d' \approx \frac{\Delta F}{F_0(1 + 1/\mathrm{SBR})} \sqrt{\frac{F_0(1 + 1/\mathrm{SBR})\tau_{1/e}}{2}} = \frac{1}{\sqrt{1 + 1/\mathrm{SBR}}} \frac{\Delta F}{F} \sqrt{\frac{F_0\tau_{1/e}}{2}} \qquad (2)$$

For the same calcium sensitivity ($\Delta F/F$) and baseline cell brightness ($F_0$), the discriminability $d'$ of calcium transients is reduced by a factor of $(1 + 1/\mathrm{SBR})^{1/2}$ in the presence of the background. For example, when $\mathrm{SBR} = 1$, the $d'$ value is reduced by a factor of 1.4, resulting in a lower true-positive or higher false-positive rate for spike detection. To compensate for the reduced $d'$, $F_0$ needs to be increased by a factor of $(1 + 1/\mathrm{SBR})$, which can only be achieved by increasing the excitation pulse energy by a factor of $(1 + 1/\mathrm{SBR})^{1/2}$ (since 2PE fluorescence $\propto P^2$) or increasing the dwell time on each neuron by a factor of $(1 + 1/\mathrm{SBR})$. Consequently, the cross-over depth between 1320-nm 3PM and 920-nm 2PM for achieving the same $d'$ with the same excitation pulse energy shifts to a shallower depth (~ 600 μm) than that based only on the signal strength (~ 700 μm), see *Figure 1B* and *Figure 2—figure supplement 2*. The SBR of 2PM drops even faster with depth than the exponential decay of the fluorescence signal because, in addition to the decrease of the signal, the background also increases with depth (*Figure 2—figure supplement 2*; *Theer et al., 2003*). Therefore, 2PM imaging far beyond the depth where $\mathrm{SBR} = 1$ is impractical.

## Tissue heating and excitation saturation limit the average and peak power

Laser-induced tissue damage can be categorized into thermal and nonlinear damage, with distinct mechanisms and dependence on imaging parameters. Continuous heating damages the brain tissue through high temperature, which disturbs various biophysical processes (*Podgorski and Ranganathan, 2016*). Heating-induced damage happens in the bulk tissue and depends on the average power per illumination volume. On the other hand, nonlinear damage is caused by the strong electric field at the focal point, which is related to focal spot size, pulse energy, and pulse duration.

We used Monte Carlo method to calculate light intensity distribution and temperature rise caused by 1320 nm illumination and compared the simulation to immunohistochemistry results, following previous works on 2PM brain heating assessment. The previous works showed that brain tissue damage was observed at >250 mW average power of continuous 920 nm illumination in both anesthetized and awake mice with cranial windows (NA = 0.8; Linear FOV = 1 mm; focal depth = 250 μm) (*Podgorski and Ranganathan, 2016*). In comparison, 1320 nm light experiences stronger water absorption and weaker scattering, which implies higher heat generation in a smaller volume, and therefore tissue temperature is expected to rise faster with input power (*Table 1*). According to our simulation, at 1 mm imaging depth, the maximum brain temperature starts to rise above the normal body temperature of 37 °C at 80 mW average power immediately after the objective lens of 2 mm

**Table 1.** Optical parameters of gray matter.

| Variable | Parameter | Wavelength | Value | Units |
|---|---|---|---|---|
| $\mu_a$ | absorption coefficient | 920 | 0.039 | 1/mm |
| | | 1280 | 0.078 | 1/mm |
| | | 1320 | 0.12 | 1/mm |
| $\mu_s$ | scattering coefficient | 920 | 6.7 | 1/mm |
| | | 1280 | 3.2 | 1/mm |
| | | 1320 | 3.2 | 1/mm |
| $g$ | anisotropy coefficient | All | 0.9 | dimensionless |
| $n$ | refractive index | All | 1.36 | dimensionless |

work distance (or 68 mW at the brain surface) with continuous scanning (*Figure 3B and C*; NA ~ 0.75; FOV = 230 μm). For power higher than 80 mW, the peak temperature rises at a rate of ~3 °C per 50 mW after the objective lens (*Figure 3C*). At low imaging power, light absorption does not raise the maximum tissue temperature but makes up for part of the heat loss through the cranial window (*Podgorski and Ranganathan, 2016*). As the input power keeps increasing, the temperature rises almost linearly with input power. The reason to distinguish the average power after the objective lens from that at the brain surface is that the absorption of 1320 nm light by immersion water is not negligible. For objective lenses with different work distances, the average power at the brain surface needs to be re-calibrated according to the actual thickness of immersion water (*Figure 3—figure supplement 1*-Materials and methods).

To experimentally determine the tissue heating at various average power, we immunolabeled *post mortem* brain slices of mice after the exposure to 20 min continuous scanning with 1320 nm 3PM at two imaging depths (1 mm and 1.2 mm, *Figure 3D*). According to the immunohistochemistry results, no tissue response was detected by measuring heat shock protein ( anti HSP-70/72), microglial (anti-GFAP), or astrocytic activation ( anti-Iba1) at 100 mW average power in anesthetized mice (NA = 0.75; Linear FOV = 230 μm; n = 5 mice, *Figure 3—figure supplement 5*); however, there is a non-zero chance (1 in 4 mice at 1 mm depth and 1 in 3 mice at 1.2 mm depth) of detectable tissue response with 150 mW imaging power, based on visual inspection of the brain slices (*Figure 3D*) and the quantification of immunolabeling intensity (*Figure 3E*). The variation in the levels of activation measured at the same average power in different mice may result from several factors, including the variation in attenuation length and the level of tissue growth 3 weeks after window implantation. Relating to the temperature calculation in *Figure 3C*, the immunohistochemistry results at 150 mW illumination suggest the upper bound of peak temperature should not exceed 41 °C, which is the peak temperature at an illumination power of 150 mW, according to our simulation (*Figure 3C*). Based on this estimated criterion, the maximum allowable imaging power can be interpolated for different imaging depths and FOVs, according to *Figure 3—figure supplement 3* and *Figure 3—figure supplement 4*. In general, the maximum brain temperature is lower with larger imaging depth or scanning FOV for the same excitation power. Brain heating is independent of the imaging frame rate when the temperature change between successive frames is negligible. For example, because the brain tissue cools at the rate of ~0.1 °C/s (*Podgorski and Ranganathan, 2016*), the temperature fluctuation at a given point in the tissue due to beam scanning is only ~0.05 °C at 2 Hz frame rate. Under this condition, the brain heating can be modeled as under continuous wide-field illumination.

In addition to 1320 nm, we also simulated brain heating with 1280 nm excitation light since the water absorption coefficient at 1280 nm is ~65% of that at 1320 nm (*Table 1*), leading to less than 50% of the total photons absorbed in the illuminated volume (*Table 2*). As a result, 3PE with 1280 nm allows almost 50% more average power than 1320 nm with similar EAL (*Kobat et al., 2009*), which is potentially beneficial for maximizing imaging depth or FOV. However, the GCaMP6 sensitivity ($\Delta F/F$) with 1280 nm 3PE in vivo is only half of that with 1320 nm, which favors 1320 nm for 3-photon calcium imaging (*Ouzounov et al., 2019*).

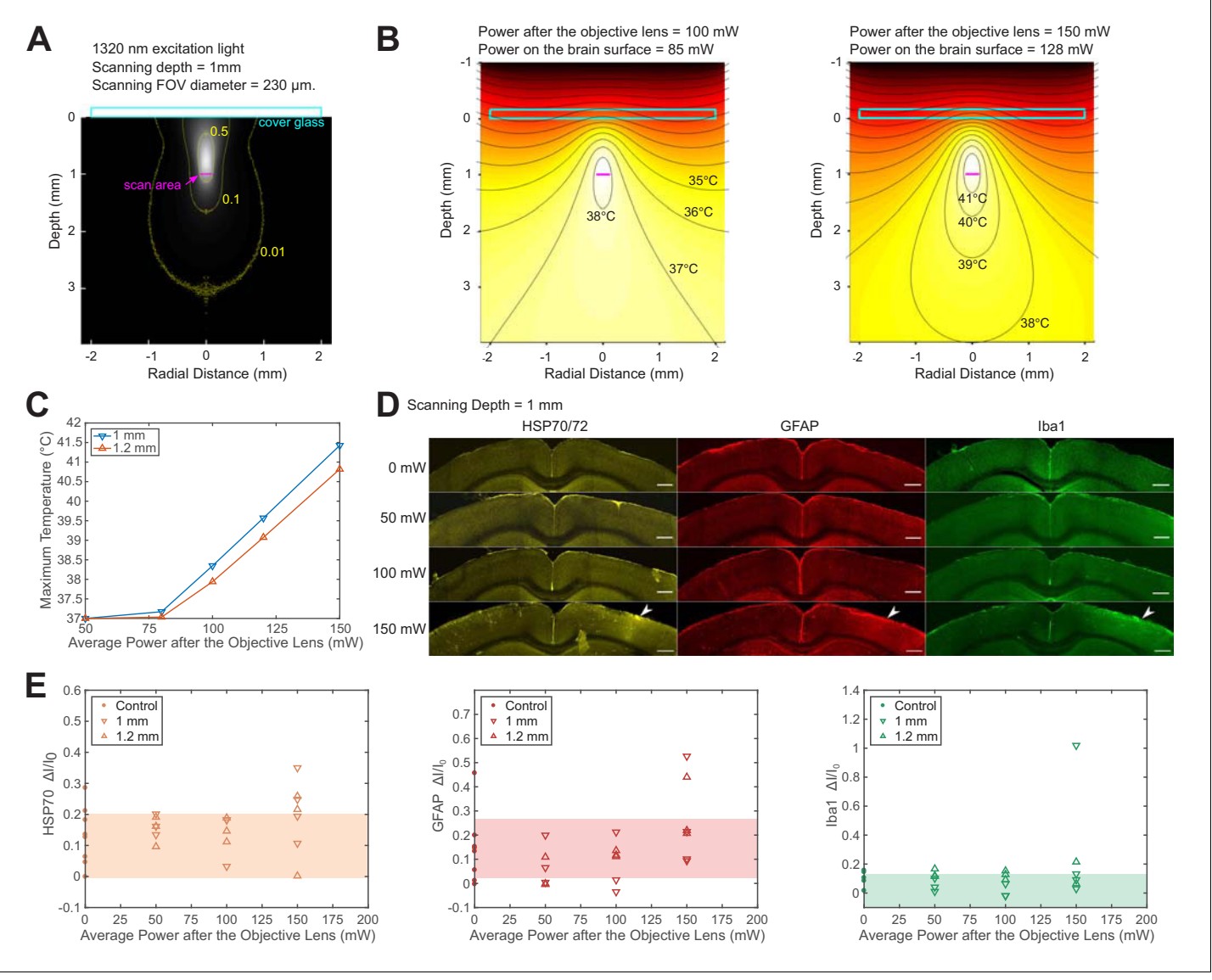

**Figure 3.** Brain heating and thermal damage induced by continuous scanning by 1320 nm 3PM. (**A**) Monte Carlo simulation of light intensity of 1320 nm excitation light. The excitation light is focused at 1 mm below the brain surface in the cortex by an objective of 1.05 NA at ~75% filling of its back aperture and scanned telecentrically in a 230 μm diameter FOV (the horizontal magenta line segment). Three iso-contour lines (from inside to outside) correspond to 0.01, 0.1, and 0.5 of the maximum intensity. (**B**) Temperature maps under 1320 nm illumination calculated from the power absorption per unit volume in (**A**) after 60 s of continuous scanning with 100 and 150 mW average power after the objective lens. Temperature is color-coded with isotherms plotted at 1˚C increment with the highest four temperature levels labeled. The average imaging power is listed at the top of each plot. (**C**) The maximum temperature versus imaging power for 1 mm and 1.2 mm focal depth, with other imaging parameters the same as in (**A**) and (**B**). The maximum temperature is calculated as the average in the hottest volume around the scanned area (~$10^7$ μm³). (**D**) Immunolabeled brain slices of mouse brains after 1320 nm 3PM scanning with 0, 50 mW, 100 mW, and 150 mW average power (one mouse is shown for each power). The brain was scanned for 20 min continuously at 1 mm below the surface, with 230 μm x 230 μm FOV and 2 Hz frame rate. The location of the damage is indicated by the white arrowheads. Scale bar, 0.5 mm. (**E**) Quantification of heat-induced damage by the fractional change of immunolabeling intensity relative to the region in the contralateral hemisphere. Shaded areas denote 95% confidence interval of the control group mean (n = 4 mice for control; n = 6 mice for 50 mW; n = 5 mice for 100 mW; n = 7 mice for 150 mW).

The online version of this article includes the following source data and figure supplement(s) for figure 3:

**Source data 1.** Quantification of the staining intensity of immunolabeld mouse brain slices *post mortem* after the exposrue to continuous 1320 nm 3PM scanning, plotted in *Figure 3E*.

**Figure supplement 1.** Light attenuation by immersion water for various excitation wavelengths.

**Figure supplement 1—source data 1.** Power transmission through immersion water of different thicknesses under the objective lens, measured with different excitation spectra and plotted in *Figure 3—figure supplement 1*.

*Figure 3 continued*

**Figure supplement 2.** Spatial dimension, coordinate system, and boundary conditions for Monte Carlo and heat conduction simulation.

**Figure supplement 3.** The maximum brain temperature as a function of the average power at the brain surface, shown for different imaging depths at 920 nm, 1320 nm, and 1280 nm.

**Figure supplement 4.** The maximum brain temperature decreases with the scanning field-of-view.

**Figure supplement 5.** Immunostaining reveals brain tissue damage by 1320 nm 3PM with 150 mW imaging power after the objective lens at 1.2 mm imaging depth.

High peak excitation intensity has several adverse effects on brain imaging, including fluorophore saturation, bleaching, and nonlinear tissue damage. For GCaMP6s, we calculated that fluorescence saturation (i.e., ground-state depletion) occurs at 4-5 nJ under 3PE under our imaging conditions (Materials and methods). Laser-induced nonlinear damage can be observed in cells as a sudden but irreversible elevation to extraordinary brightness, which is caused by the rapid ionization and recombination of molecules followed by pressure increase (*Tsai et al., 2009*). In general, the damage threshold for the peak intensity increases with wavelength (*Fu et al., 2006*; *König et al., 1999*; *Olivié et al., 2008*). Since the area of the focal spot scales as $\lambda^2$, where $\lambda$ is the excitation wavelength, the long wavelength used for 3PM allows higher pulse energy than 2PM. Our previous work showed that neurons remain healthy and viable for weeks after hours of exposure to 1.5 nJ pulses (60 fs and NA~0.75) at the 1300-nm wavelength (*Ouzounov et al., 2017*). As an estimate of the upper bound of pulse energy, we observed that ~10 nJ at the focus causes tissue ablation, in agreement with another independent study (*Yildirim et al., 2019*). In addition, it was shown that 1 to 2 nJ pulse energy at the focus (~ 40 fs, NA ~ 1.0) does not appear to alter the physiological response of the neurons under visual stimulation (*Yildirim et al., 2019*). Based on these results, 1 to 2 nJ pulse energy at the focus is reasonable for 3PM to achieve sufficient signal strength while avoiding nonlinear photodamage and fluorophore saturation.

## Discussion

In this study, we quantitatively compared 1320 nm 3PM and 920 nm 2PM for deep tissue GCaMP6 imaging. The 1320 nm 3PM has the benefit of more efficient signal generation and substantially higher SBR for calcium imaging in the deep mouse brain cortex. We find that 1320 nm 3PM outperforms 2PM in signal generation efficiency at a depth beyond ~700 μm in mouse neocortex regardless of labeling density. In the densely labeled mouse brain, for example CaMKII-GCaMP6s transgenic mouse, 1320 nm 3PM is preferred beyond the depth where SBR ~ 1 (e.g.,~4 EALs or 600 μm in mice neocortex). In practice, depending on the labeling density and expression nonuniformity, 3PM might be considered at an even shallower depth to prevent recording bias against neurons with lower expression levels, for example ~3 EALs or ~450 μm (*Figure 3D*).

We formulate a step-by-step procedure to optimize 3PM for neuronal imaging using the results in this paper. The excitation parameters (wavelength, pulse energy, and repetition rate, etc.) are optimized first, based on which the sampling parameters (FOV, frame rate, pixel size, etc.) are then derived accordingly.

**Table 2.** Percentage of Excitation Photons for Various Final Destinations.

|  | 920 nm | 1320 nm | 1280 nm |
|---|---|---|---|
| Contributing to heating | 20 | 63 | 49 |
| Back scattered to window | 25 | 9 | 10 |
| Back scattered to skull | 20 | 9 | 11 |
| Escaped* | 35 | 19 | 30 |

*Photons escape the simulation volume by traveling too far (>6 mm) from the center of the cranial window or too deep (>6 mm) from the tissue surface.

The excitation wavelength affects 3PE cross section, calcium indicator $\Delta F/F$, and tissue heating. A previous study has shown that 3PE of GCaMP6s achieves the highest $d'$ with a center wavelength between 1300-1320 nm (*Ouzounov et al., 2019*). Brain heating is significantly increased beyond 1350 nm due to water absorption (*Jacques, 2013*) and much reduced at 1280 nm (see the previous section). Even though 1280 nm allows ~ 50% more average power than 1320 nm, it only generates ~50% more signal since the signal scales linearly with the average power when the peak intensity is maximized and fixed. For calcium imaging, this advantage in signal strength is offset by the lower $\Delta F/F$, which results in a slightly lower $d'$ for 1280 nm (*Ouzounov et al., 2019*).

The repetition rate and pulse energy are optimized based on the constraints imposed by the linear (brain heating) and nonlinear effects. Since 3-photon imaging is essentially background-free for most practical imaging depths, the calcium imaging fidelity $d'$ depends solely on the signal strength, that is photons per neuron per second ($F_0$). As a result, in order to maximize the number of neurons recorded, the total photon counts should be maximized within the thermal constraint and the peak intensity limit. Regardless of the imaging depth, the pulse energy at the focus should always be kept as high as possible (e.g., 1 to 2 nJ) to maximize the signal while avoiding the adverse nonlinear effects (e.g., fluorophore saturation and nonlinear damage). The maximum repetition rate is then the maximum average power allowed by the thermal constraint (e.g., as indicated in *Figure 3—figure supplement 3*) divided by the pulse energy at the sample surface, which can be calculated from the pulse energy at the focus using the EAL. The imaging power grows exponentially with depth, which is much faster than the quasi-linear growth of the heat dissipation capacity as a function imaging depth (*Figure 3—figure supplement 3*). Therefore, the maximum repetition rate decreases rapidly with imaging depth. For example, when imaging GCaMP6s-labeled neurons at 600 μm depth (~ 2 EALs) using 1320 nm 3PE, 1.86 nJ x exp(2) ~ 14 nJ pulse energy on the brain surface is required for 0.1 photon/pulse detected (*Figure 1B*), and the maximum repetition rate to avoid substantial tissue temperature rise is 100 mW/ 14 nJ = 7 MHz, where 100 mW is predicted as a safe power to avoid thermal damage at this imaging depth, given a large enough scanning FOV (*Figure 3—figure supplement 3* and *Figure 3—figure supplement 4*). When imaging the hippocampus at ~ 1 mm depth (~ 4 EALs due to the presence of the white matter), the maximum repetition rate has to be reduced to 120 mW/100 nJ = 1.2 MHz for the same signal generation level at the focus. In both examples, the pulse energy at the brain surface was calculated by multiplying exp(the number of EALs) with the pulse energy at the focus (1.86 nJ).

The repetition rate fundamentally limits the number of samples per second, and therefore the possible range of FOVs and frame rate. As a common practice, the pixel size is approximately the lateral spot size of the focus, for example two pixels per focal spot for a proper spatial sampling frequency. To ensure sampling uniformity when the repetition rate is low, the pixel clock is synchronized to the laser pulses, and there is an integer number of excitation pulses per pixel. Under this condition, a wide range of FOVs and frame rates can be chosen as long as the repetition rate is equal to the integer multiples of the product of the number of pixels in each frame and the frame rate. In addition, the following factors should also be taken into consideration for parameter selection. Tissue heating is higher with a smaller FOV, and therefore the imaging power should be reduced accordingly with the FOV (see *Figure 3—figure supplement 4*). For calcium imaging, the frame rate should be fast enough for capturing the temporal dynamics of the calcium indicator (e.g., >5 Hz for GCaMP6s).

The maximum repetition rate limits the sampling rate for 3PM, and therefore the maximum number of neurons can be recorded per unit time. On the other hand, the excitation pulse train does not have to be periodic in time. Further optimization by on-demand delivery of pulses to the ROIs only, for example by using acousto-optical deflectors with proper gating of the laser pulses (*Duemani Reddy et al., 2008*; *Grewe et al., 2010*; *Katona et al., 2012*) or an adaptive excitation source (*Li et al., 2020*), can improve the imaging speed or increase the number of neurons recorded per unit time by at least an order of magnitude. Our discussion on the optimization of the imaging parameters is still valid for adaptive excitation of the ROIs only, except that the time-averaged 'repetition rate', that is the average number of pulses per second, should be used instead of the conventional repetition rate.

# Materials and methods

**Key resources table**

| Reagent type (species) or resource | Designation | Source or reference | Identifiers | Additional information |
|---|---|---|---|---|
| Strain, strain background (*Mus musculus*) | B6.Cg-Tg(CamK2a-tTA)1Mmay/J | The Jackson Laborartory Stock: 007004 | MGI:2179066 | |
| Strain, strain background (*Mus musculus*) | B6;DBA-Tg(tetO-GCaMP6s)2Niell/J | The Jackson Laborartory Stock: 024742 | MGI:5553332 | |
| Antibody | anti-HSP70/72 (mouse monoclonal) | Enzo Life Sciences, Cat# SPA-810PED | RRID:AB-2264369 | IHC(1:400) |
| Antibody | anti-GFAP (mouse monoclonal) | Sigma-Aldrich, Cat# G3893 | RRID:AB_477010 | IHC(1:760) |
| Antibody | anti-Iba1 (mouse monoclonal) | Sigma-Aldrich, Cat# SAB2702364 | RRID:AB_2820253 | IHC(1:1000) |
| Antibody | Goat anti-mouse (polyclonal) | Thermo Fisher Scientific, Cat# A-11003 | RRID:AB_2534071 | IHC(1:500) |
| Software, algorithm | *Source code 1*: matlab code for simulating the brain temperature distribution under continuous long-wavelength illumination by 3PM using Monte Carlo method and heat equation. | Mathworks, Matlab 2016b | RRID:SCR_001622 | |

## Simultaneous 1320 nm 3PM and 920 nm 2PM Imaging with Time Division Multiplex Scheme

The excitation source for 1320 nm 3PM was a noncollinear optical parametric amplifier (NOPA, Spectra-Physics) pumped by an ultrafast amplifier (Spirit, Spectra-Physics). The excitation source for 920 nm 2PM was a mode-locked Ti:Sapphire laser (Tsunami, Spectra-Physics). The two excitation beams were launched into a custom-built microscope, described in our previous works (*Ouzounov et al., 2019*; *Ouzounov et al., 2017*). The 3PE wavelength was centered at 1320 nm and 2PE at 920 nm for the optimal $d'$ (*Ouzounov et al., 2019*). The two excitation beams were verified to have similar spatial resolution well below the size of neurons or blood vessels such that their excitation volumes are almost identical (*Ouzounov et al., 2019*).

Time-division multiplex (TDM) achieves nearly simultaneous 2PM and 3PM imaging by alternating between spatially overlapped 920 nm and 1320 nm laser beams on a microsecond time scale. 2PE and 3PE fluorescence signals were separated according to the recorded laser clock. More details about the setup can be found in *Ouzounov et al. (2017)*. The calcium activities were recorded with 13.6 Hz frame rate, limited by the fastest achievable line rate of the galvanometer scanners. This frame rate is sufficiently high for imaging the temporal dynamics of GCaMP6s ($\tau_{1/e} \sim 2$ s). For 2PM and 3PM comparison experiments, the FOV was reduced to fit a single neuron so that the signal-to-noise ratio was maximized to ensure accurate comparison between 2PM and 3PM calcium traces. The excitation power for 3PM and 2PM was adjusted to generate similar photon count per frame that lead to the same noise level, and the FOV was reduced to increase the laser dwell time on neuron bodies.

## Photon counting schemes

The microscope was first tested for shot-noise limited performance by photon counting the signal generated by stationary 1320-nm or 920-nm beam focused in fluorescein solution (~40 μM and pH=10). The laser power was chosen (~0.3 mW for both wavelengths) to ensure the photon counts per second is lower than 5% of the laser repetition rate of the NOPA, which limits the photon stacking error to within 2.5% of the total counts. According to Poisson statistics, photon stacking error causes a fraction of $1 - (1 - e^{-\lambda})/\lambda$ underestimation in photon counts, where $\lambda$ is the average count per second divided by the laser repetition rate. For 2PE using the Ti:Sapphire laser, the photon

stacking error is negligible since the typical photon counts are below 1% of its repetition rate (80 MHz). In this experiment, the PMT (Hamamatsu H7422-40) anode current was first amplified with a 10 MHz bandwidth pre-amplifier (C9999, Hamamatsu), and then counted by a photon counter (SR400, Stanford Research). Shot-noise limited performance was confirmed for our imaging system.

During imaging, the photon counts of neurons were obtained by converting pixel values to photon counts according to a conversion factor. The calibration was done by parking a 920 nm or 1320 nm focus in a fluorescein solution sample and then perform photon counting and imaging consecutively. Linearity between pixel values and photon counts was tested by changing the laser power, and the ratio between them was used as the conversion factor. In fact, recording analog value is a better way to measure high photon counts, since there is no photon stacking error and the voltage signal can simply be summed. The conversion factor was further confirmed by observing the first mode in a pixel value histogram. The zeroth mode of a pixel histogram peaks at PMT offset value, and the first mode is larger than that, representing pixels receiving exactly one photon. Higher-order modes can also be observed, representing pixels receiving multiple photons. For all simultaneous 2P and 3P imaging, the sampling frequency was 5 MHz, so that direct photon counting based on images could also be performed, which showed consistent results in comparison to analog recordings.

## Measurement of excitation light attenuation in the brain tissue based on fluorescence signal

The vasculature imaging was performed simultaneously with 2PM and 3PM using the TDM scheme. The image stack was taken with 10 μm step size in depth, and the imaging power was increased with imaging depth to keep the signal level approximately constant. The signal of each frame was calculated as the average of the brightest 0.5% pixel values and then converted to photon counts per excitation pulse according to the method described in the previous section. The fraction of excitation power reaching the focus from the brain surface (Focus Power/Surface Power) was calculated as the square (cubic) root of the ratio of the 2PE (3PE) signal at the imaging depth and at the brain surface.

## Measurement of the pulse energy required per 0.1 photon detection

To have a fair comparison on the excitation efficiency of 920 nm 2PM and 1320 nm 3PM, we controlled all the parameters regarding 2PE and 3PE except for the wavelength. For both wavelengths, the axial resolution was made equal by adjusting the beam size at the back aperture of the objective separately. The point spread functions of both wavelengths were also overlapped laterally (by fine-tuning the 920 nm beam pointing direction) and axially (by changing the field curvature of the 920 nm beam at the objective lens back aperture by fineadjustment of the distance between the lenses of a 1:1 telescope). For both wavelengths, the pulse durations were scaled to 60 fs (measured pulse durations were 60–70 fs in FWHM for 1320 nm, and 180–200 fs for 920 nm. All pulses were assumed to have sech$^2$ profile). Light absorption by the immersion water was taken into account when calculating the pulse energy for 1320 nm on the brain surface. As we show in *Figure 3—figure supplement 1*, it is important to distinguish the pulse energy on the brain surface and after the objective lens for 1320 nm excitation.

Based on the fluorescence signals derived from simultaneous 2P and 3P imaging, *Figure 1B* was plotted using the following equations (*Xu and Webb, 1997*):

$$S_{2P}/f = C_2(P/f)^2/\tau \tag{3}$$

$$S_{3P}/f = C_3(P/f)^3/\tau^2 \tag{4}$$

where $S_{nP}/f$ is the n-photon-excited signal yield in the unit of detected photon/pulse, $P$ is the average power at the brain surface, $f$ is the repetition rate, $\tau$ is the pulse duration, and $C_n$ (n=2 and 3) are the coefficients to be determined. For vasculature imaging, $S_{nP}/f$ is calculated as the average of the brightest 0.5% pixel values per frame, converted to photon counts. For GCaMP6s-labeled neurons, $S_{nP}/f$ was calculated from the sum of time-averaged pixel values in cell bodies (averaging time = 75 s, on 37 different cells in 5 animals). Given that $P/f$ and $\tau$ are already measured during the experiment, $C_n$ can be solved. To produce *Figure 1B*, we plugged $C_n$ into the equations and set the

signal yield (i.e., $S_{\mathrm{nP}}/f$) to 0.1 photon per pulse, pulse duration to 60 fs, and then solve for pulse energy $P/f$ on the sample surface according to *Equations 3 and 4*.

To account for the EAL variation recorded from different animals in *Figure 1B*, we measured EAL of the cortical layers of each animal by imaging fluorescein-labeled blood vessels immediately after calcium imaging. The depths of neurons were rescaled by dividing the measured EAL and then multiplying with the nominal EAL (i.e., 293 µm for 1320 nm 3PE and 154 µm for 920 nm, as measured by the vasculature data). The logarithm of the neuron signal was plotted against depth, and then fitted with a linear model for the slope and intercept (*Figure 1B*). The slope equals $ln(10)/\mathrm{EAL}$, and the intercept equals to the pulse energy required to yield 0.1 detected photon per pulse on the sample surface (i.e., zero imaging depth).

### Measurement of Signal-to-background ratio

The signal of each frame was measured as the average of the brightest 0.1% pixel values. For vasculature data, the background was measured as the average pixel values of regions surrounding the blood vessels (i.e., not labeled). For GCaMP6 neural data, the background was measured as the pixel value in the shadow of big blood vessels. The pixel values of cortical tissue surrounding the neurons cannot be used as background since it contains densely labeled neuronal processes except for area within the blood vessels. SBR was calculated as the signal divided by the background.

### Quantification of vasculature volume fraction and staining inhomogeneity

The SBR limit of 2PM depends on the spatial inhomogeneity of staining (*Theer and Denk, 2006*). We quantified the staining inhomogeneity of fluorescein-labeled vasculature in order to compare the in vivo SBR measurement to the theory and ex vivo fluorescent bead measurement (*Theer and Denk, 2006*). The volume fraction of the vasculature can be estimated by the fraction of the stained blood vessel area in each xy image frame. *Figure 2—figure supplement 1* shows the fractional vascular area vs. imaging depth, derived from the same 3PM dataset, as in *Figure 2B*. The segmentation of blood vessel regions was performed with *graythresh* function in MATLAB and then inspected manually for correctness. We concluded that the labeled vascular volume accounts for 2 ± 1% (mean ± standard deviation) in the imaged column of the mouse cortex (~2mm lateral and 2mm caudal to the Bregma point, *Figure 2B*). Our result is in close agreement with other studies quantifying the fractional vascular volume by imaging sliced brains ex vivo (~2% for blood vessels of 20 µm or less diameter in the primary somatosensory cortex) (*Wang et al., 2019*; *Xiong et al., 2017*). We noticed that most of the blood vessels in the imaged volume has a diameter of less than 20 µm, except for a few on the brain surface (*Figure 2—figure supplement 1A*). The blood vessel density is lower in the white matter (~750-850 µm) than that in the cortex (*Figure 2—figure supplement 1B*). The staining inhomogeneity is defined as, $\chi = \hat{C}/\langle C \rangle$, where $C = C(x, y, z)$ denotes the spatial distribution of dye concentration in the entire imaged volume, $\hat{C}$ is the maximum concentration, and $\langle C \rangle$ is the average concentration (*Theer and Denk, 2006*). Staining inhomogeneity can be estimated as 1/(factional vascular volume) with the assumption that all the labeled blood vessels are equally bright ($C = 1$ for all vasculature), and the rest of tissue is completely unstained ($C = 0$ for the rest). Our experiments show that 2PM reaches SBR of 1 at about 4.7 EALs, that is 730 µm with EAL=154 µm (*Figure 2B*). This result is in close agreement to the 4.7 EALs predicted by theoretical calculation with a staining inhomogeneity of 50 (*Theer and Denk, 2006*).

### Analysis of the calcium traces of neurons

Sample motion in the original images, if any, was corrected by TurboReg plug-in in ImageJ. Regions of interest (ROIs) were generated by manual segmentation of neuron bodies. The pixel values of ROIs were exported to MATLAB 2016b for further processing. All the pixels in the ROI were summed and converted to photon counts. Spikes were inferred by thresholding the Poisson-distribution-based likelihood function derived from each trace (*Wilt et al., 2013*). The discrimination threshold (denoted by C in the cited paper) was chosen to be $ln((1 - r)/r)$, where $r$ is the estimated firing rate, which was estimated for each individual trace as the fraction of the trace that is more than 1.5 standard deviation above its mean. The baseline ($F_0$) was determined by averaging trace values, after excluding the spikes and their rising and falling edges. For the processed traces in *Figure 1C*,

fluorescence intensity traces were low-pass filtered with a hamming window of a time constant of 0.37 s. Traces ($F$) were normalized according to the equation $(F - F_0)/F_0$ .

## Measurement of $\Delta F/F$ ratio from Simultaneously Recorded 3PM and 2PM Calcium Traces

Based on the low-pass-filtered and normalized calcium traces described in the section above, the peaks of spikes were detected by finding local maxima that are larger than 30% $\Delta F/F$ in 3PM traces and have corresponding spike peaks detectable in 2PM traces. To produce the plot of *Figure 2D*, the ratios of 2PM and 3PM $\Delta F/F$, that is $(\Delta F/F)_{2P}/(\Delta F/F)_{3P}$, of the same calcium transients were taken, and the depth of each neuron was normalized with the EAL of each animal.

## Monte Carlo simulation of light propagation and validation of tissue optical parameters

We used Monte Carlo simulation to calculate light propagation in the brain tissue, following the algorithm described previously (*Stujenske et al., 2015*). Podgorski et al. adapted the same numerical recipe to predict tissue temperature change during 2-photon imaging, which agreed well with experimental measurements at the wavelengths of 800 nm, 920 nm, and 1064 nm (*Podgorski and Ranganathan, 2016*). In order to simulate for 1320 nm and 1280 nm excitation, we measured and estimated brain optical parameters as follows: For excitation wavelength longer than 1200 nm, water accounts for the majority of the light absorption in brain tissue in vivo (*Jacques, 2013*). Therefore, tissue absorption coefficients $\mu_a$ was approximated with 75% of the spectrum-weighted water absorption coefficient (defined in Section Measurement of Optical Absorption by Immersion Water), and the 75% is based on the water content of brain tissue (*Jacques, 2013*; *Tschöp et al., 2012*). Tissue scattering coefficients $\mu_s$ were then calculated from our measured effective attenuation lengths according to $\text{EAL} = 1/(\mu_a + \mu_s)$, with EAL=150 µm at 920 nm, and 300 µm at 1320 nm and 1280 nm. As cross-validation, the calculated scattering coefficient at 1320 nm is close to that measured by Gebhart et al. (i.e., 3.0 mm$^{-1}$) (*Gebhart et al., 2006*). We assumed an anisotropy factor $g = 0.9$ and tissue refractive index $n = 1.36$ for 920 nm, 1280 nm, and 1320 nm, since their variation in the wavelength range of interest is negligible. All the tissue optical parameters are summarized in *Table 1*.

To simulate underfilling of the objective back aperture, we initialized random photon distribution incident to the brain surface according to the relations:

$$w = w_0 \sqrt{-1/2\ln(X)}$$
$$\theta = sin^{-1}(w/n_0 f)$$
$$r = tan(\theta)z$$

$$(5)$$

where $X$ is a random variable drawn from a uniform distribution from 0 to 1, $w_0$ is the 1/e$^2$ beam radius of the Gaussian beam at the objective back aperture, $w$ is the radial distance to the objective optical axis of a randomly generated photon conforming to the Gaussian beam profile mentioned above, $f$ is the objective focal length, $n_0$ is the refractive index of immersion water, $z$ is the imaging depth in the sample. $r$ and $\theta$ define, respectively, the radial coordinate of photon position and the polar angle of propagation direction, both with respect to the objective optical axis (*Figure 3—figure supplement 2*). The objective (Olympus XLPLN25XWMP2, 25X, NA=1.05, focal length=7.2 mm) is under-filled, with 70% of its back aperture matched to the 1/e$^2$ beam diameter of a Gaussian beam, which gives an effective NA of ~ 0.75 (*Theer and Denk, 2006*). The geometry of the simulation volume and boundary conditions are shown in *Figure 3—figure supplement 2*.

## Heat diffusion model with Bio-heat equation

Using the light intensity as the heat source, we calculated the temperature distribution by solving numerically the bio-heat *Equation (6)*, identical to that used previously by *Stujenske et al. (2015)*:

$$\rho c \frac{\partial T(\vec{r},t)}{\partial t} = k\nabla^2 T(\vec{r},t) + \rho_b c_b w_b (T_A - T(\vec{r},t)) + S_h(\vec{r}) + q_m \qquad (6)$$

where $T(\vec{r},t)$ is the spatial-temporal temperature distribution, $S_h(\vec{r})$ is the radiative heat generation from Monto Carlo simulation, and the rest of the parameters and their values are listed in *Table 3*, which are the same as in *Stujenske et al. (2015)*.

**Table 3.** Thermal and mechanical properties of gray matter (*Stujenske et al., 2015*).

| Variable | Parameter | Value | Units |
|---|---|---|---|
| $\rho$ | Density | $1.04 \times 10^{-3}$ | g/mm$^3$ |
| $c$ | Brain specific heat | $3.65 \times 10^3$ | mJ/g°C |
| $k$ | Thermal conductivity | 0.527 | mW/mm °C |
| $\rho_b$ | Blood density | $1.06 \times 10^{-3}$ | g/mm$^3$ |
| $c_b$ | Blood specific heat | $3.6 \times 10^3$ | mJ/g°C |
| $w_b$ | Blood perfusion rate | $8.5 \times 10^{-3}$ | /s |
| $q$ | Metabolic heat | $9.5 \times 10^{-3}$ | mW/mm$^3$ |
| $T_A$ | Arterial temperature | 36.7 | °C |

We calculated steady-state temperature distribution at various imaging depths (0–6 attenuation lengths) and average powers for 920 nm, 1320 nm, and 1280 nm. *Figure 3—figure supplement 3* shows the maximum tissue temperature as a function of the average input power on the brain surface. The maximum temperature is evaluated as the average temperature in a volume of ~$10^7$ µm$^3$ (*Figure 3—figure supplement 3*), or equivalent to a cylinder of ~120 µm radius and ~210 µm height enclosing the hottest region of the tissue. The power at the brain surface is calculated from the power immediately after the objective lens by taking account of the absorption by immersion water.

## Immunohistolochemistry for assessing thermal damage induced by 1320 nm Illumination

The experiment was performed 3 weeks after the window implantation. Anesthetized (2% isoflurane mixed with oxygen) mice were exposed to continuous scanning for 20 min at a 2-Hz frame rate and 230 µm x 230 µm FOV. The FOV was chosen to be similar to the actual achievable FOVs at the imaging depth of 1-1.2 mm (*Ouzounov et al., 2017*; *Weisenburger et al., 2019*), and the frame rate was chosen to be comparable to that in typical structural imaging. Even at 2 Hz, the frame rate is fast enough to neglect tissue cooling (~0.1 °C/s) between successive frames (*Podgorski and Ranganathan, 2016*). After 18 hours, mice were perfused with 4% paraformaldehyde, and post fixed in the same solution for 24 hours, followed by sucrose solutions immersion (10%, 20%, and 30% incrementally). Scanned brain regions were coronally sectioned into 200 µm sections and blocked at room temperature. Alternating slices were incubated at 4 °C overnight, with primary antibodies for heat shock protein (HSP70/72), glial fibrillary acidic protein (GFAP), and Iba1. Primary antibodies used were mouse monoclonal anti-HSP70/72 (C92F3A-5) (1: 400 dilution, Enzo Life Sciences Cat# SPA-810PED, RRID:AB-2264369), mouse anti-GFAP (1:760 dilution, Sigma-Aldrich Cat# G3893, RRID:AB_477010), and mouse anti-Iba1 (1: 1000 dilution, Sigma-Aldrich Cat# SAB2702364, RRID:AB_2820253). Slices were washed and incubated with secondary antibody conjugated to Alexa Fluor 546 (1:500 dilution, Thermo Fisher Scientific Cat# A-11003, RRID:AB_2534071) at 4 °C overnight, washed again, and mounted in VECTASHIELD antifade medium for confocal imaging. Images were analyzed with ImageJ. The intensity ($I_0$) was determined by the average intensity in the region of interest ~1 mm wide centered around the illuminated site covering the entire thickness of the neocortex, and the baseline $I_0$ was determined in the mirror position in the contralateral hemisphere. The level of tissue response was quantified as the fractional change of immunolabeling intensity relative to the region of contralateral hemisphere: $(I - I_0)/I_0$. The procedures described here follows closely the previous studies on assessing thermal damage caused by 2-photon imaging of mouse brain (*Podgorski and Ranganathan, 2016*).

## Calculation of the pulse energy for fluorophore saturation under Three-photon excitation

The excitation probability per pulse for a fluorescent molecule at the center of the focus is calculated as (*Xu and Webb, 1997*):

$$Pr = 1 - \exp\left[-\frac{g_p^{(3)}}{\tau^2}\sigma_3\left(\frac{NA^2\pi}{\lambda_{3P}^2}\right)^3\left(\frac{P_{3P}}{f}\right)^3\right] \tag{7}$$

where the definition and values of all the parameters remain the same as in the previous section. Using GCaMP6s 3PE cross section of $3 \times 10^{-82}$ cm$^6$s$^2$, the pulse energy (i.e., $\frac{P_{3P}}{f}\cdot\frac{hc}{\lambda_{3P}}$) required for 10% and 63% excitation probability per pulse is 2 nJ and 4.3 nJ, respectively. We note that these pulse energies are much smaller (by ~ $2\pi$) than those given by *Yildirim et al. (2019)*. This factor of $2\pi$ error in *Yildirim et al. (2019)* is probably due to a typo (the Planck's constant $h$ and $\hbar$) since the correct equation for 2PE saturation intensity was given by the same authors in earlier work (*So et al., 2000*).

## Animal procedures

Chronic craniotomy was performed on mice according to the procedures described in *Ouzounov et al. (2017)*. Windows of 5 mm diameter were centered at ~2.5 mm lateral and ~2 mm caudal from the bregma point over the somatosensory cortex. Vasculature imaging was performed on wild-type mice (8–15 weeks, male, C57BL/6J, The Jackson Laboratory) with retro-orbital injection of fluorescein dextran conjugate (10 kDa, 25 mg dissolved in 200 µl saline). Calcium imaging was performed on five different transgenic animals with GCaMP6s-labeled neurons (3 males and 2 females, 11–17 weeks, CamKII-tTA/tetO-GCaMP6s). The spontaneous calcium activity imaging was performed on awake or lightly anesthetized (0.5–1% isoflurane mixed with oxygen) animals. Thermal damage experiments were performed on wild-type mice (3 to 4 mice for each depth and power combination, for a total of 23 mice, 8–30 weeks, male, C57BL/6J, The Jackson Laboratory). All animal experimentation and housing procedures were conducted in accordance with Cornell University Institutional Animal Care and Use Committee guidance.

## Acknowledgements

We thank Dr. Kaspar Podgorski for sharing the code for brain heating simulation and the protocol for immunohistochemistry. We also thank Dr. Peter So for the discussion on 3-photon excitation saturation. This work has been supported by National Science Foundation (DBI-1707312, Neuronex Hub), Intelligence Advanced Research Projects Activity (D16PC00003), National Institutes of Health (DP2MH109982), and Cornell Neurotech Mong Fellowships.

## Additional information

### Funding

| Funder | Grant reference number | Author |
| --- | --- | --- |
| National Science Foundation | DBI-1707312 | Chunyan Wu<br>Dimitre G Ouzounov<br>Fei Xia<br>Xusan Yang |
| National Institutes of Health | DP2MH109982 | Wenchao Gu |
| Intelligence Advanced Research Projects Activity | D16PC00003 | Tianyu Wang<br>Dimitre G Ouzounov |
| Cornell University | | Tianyu Wang |

The funders had no role in study design, data collection and interpretation, or the decision to submit the work for publication.

### Author contributions

Tianyu Wang, Conceptualization, Data curation, Software, Formal analysis, Validation, Investigation, Visualization, Methodology, Project administration; Chunyan Wu, Conceptualization, Resources, Data curation, Formal analysis, Validation, Investigation, Methodology; Dimitre G Ouzounov, Formal analysis, Validation, Methodology; Wenchao Gu, Resources, Methodology; Fei Xia, Minsu Kim, Xusan

Yang, Resources; Melissa R Warden, Resources, Supervision, Funding acquisition, Project administration; Chris Xu, Conceptualization, Resources, Formal analysis, Supervision, Funding acquisition, Investigation, Methodology, Project administration

**Author ORCIDs**
Tianyu Wang  https://orcid.org/0000-0002-6087-6376
Chunyan Wu  https://orcid.org/0000-0002-6294-4512
Fei Xia  http://orcid.org/0000-0001-6591-8769
Melissa R Warden  http://orcid.org/0000-0003-2240-3997
Chris Xu  https://orcid.org/0000-0002-3493-6427

**Ethics**
Animal experimentation: This study was performed in strict accordance with the recommendations in the Guide for the Care and Use of Laboratory Animals of the National Institutes of Health. All of the animals were handled according to approved institutional animal care and use committee (IACUC) protocols (#2010-0031) of Cornell University. All surgery was performed under isoflurane anesthesia, and every effort was made to minimize suffering.

**Decision letter and Author response**
Decision letter https://doi.org/10.7554/eLife.53205.sa1
Author response https://doi.org/10.7554/eLife.53205.sa2

# Additional files

**Supplementary files**
• Source code 1. Matlab code for simulating the brain temperature distribution under continuous long-wavelength illumination by 3PM using Monte Carlo method and heat equation, which was used to produce *Figure 3B and C*, *Figure 3—figure supplements 3* and *4*.

• Transparent reporting form

**Data availability**
All the parameters for calculation and models have been summarized as tables. The source data for all the figures have been provided. All the simulation codes have been uploaded and are available for download.

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

## Appendix 1

### Estimation of the Cross-over Depth Based on Fluorophore Cross Section

The number of signal photons generated per excitation pulse at the focus can be calculated according to *Equations (8) and (9)*, for 2PE and 3PE in a Gaussian beam focus respectively (*Xu and Webb, 1997*):

$$2\text{PE signal photons per pulse} = \frac{S_{2\text{P}}}{f} = \frac{1}{2}\frac{g_p^{(2)}}{\tau}\phi C(\eta\sigma_2)n\frac{\pi}{\lambda_{2\text{P}}}\left(\frac{P_{2\text{P}}}{f}\right)^2 \tag{8}$$

$$3\text{PE signal photons per pulse} = \frac{S_{3\text{P}}}{f} = \frac{1}{3}\frac{g_p^{(3)}}{\tau^2}\phi C(\eta\sigma_3)n\frac{2\pi^2}{3\lambda_{3\text{P}}^3}\text{NA}^2\left(\frac{P_{3\text{P}}}{f}\right)^3 \tag{9}$$

where $g_p^{(n)}$ is the nth-order temporal coherence factor (we assume a Gaussian temporal profile of the pulse); $\tau$ is the laser pulse width; $\phi$ is the system collection efficiency; $C$ is the concentration of the fluorophore; $\eta\sigma_n$ is the n-photon action cross section; $n$ is the refractive index of the medium; $\lambda$ is the excitation wavelength in vacuum; $\text{NA}$ is the numerical aperture defined by $1/e^2$ beam diameter at the objective back aperture; $P_{2\text{P}}$ and $P_{3\text{P}}$ are, respectively, the average power at the focus for 2PE and 3PE in the unit of photon/s, and $f$ the repetition rate.

At a typical signal yield for 3-photon imaging at 0.1 photon detected per excitation pulse (i.e. $S_{3\text{P}}/f = 0.1$), the required pulse energy at the focus (i.e. $\frac{P_{3\text{P}}}{f}\cdot\frac{hc}{\lambda_{3\text{P}}}$) has been measured to be ~2 nJ. For the 2PE signal yield to be the same as 3PE (i.e. $S_{2\text{P}}/f = 0.1$), the 2PE pulse energy at the focus can be derived from *Equations (8) and (9)*:

$$\frac{P_{2\text{P}}}{f} = \sqrt{\frac{4\pi}{9}\frac{1}{\tau}\frac{g_p^{(3)}}{g_p^{(2)}}\frac{\sigma_3}{\sigma_2}\frac{\lambda_{2\text{P}}}{\lambda_{3\text{P}}^3}\text{NA}^2\left(\frac{P_{3\text{P}}}{f}\right)^3} \tag{10}$$

For estimation, we use $10^{-49}$ cm$^4$ s/photons and $10^{-82}$ cm$^6$ (s/photons)$^2$ for 2PE and 3PE action cross sections, respectively, based on the typical values of fluorescent dyes and calcium indicators (*Cheng et al., 2014*). Evaluating *Equation 10* with all the parameter values listed in *Appendix 1—table 1*, 2PE pulse energy (i.e. $\frac{P_{2\text{P}}}{f}\cdot\frac{hc}{\lambda_{2\text{P}}}$) is solved to be 0.2 nJ. In other words, the ratio of 3PE to 2PE pulse energy resulting in 0.1 photons detected per excitation pulse is 2 nJ/ 0.2 nJ = 10 times, close to the 8x measured with GCaMP6s (*Figure 1B*). The ratio measured with fluorescein is higher (15x) because fluorescein 3PE cross section at ~ 1320 nm is noticeably smaller than $10^{-82}$ cm$^6$ (s/photons)$^2$ and 1320 nm is likely not the peak 3PE wavelength for fluorescein (*Cheng et al., 2014*).

**Appendix 1—table 1.** Two- and three-photon excitation parameters.

| | Objective lens focal length (mm) | 1/e$^2$ beam diameter at the back aperture (mm) | Effective NA | Pulse duration $\tau$ (fs) | $g_p^{(n)}$ |
|---|---|---|---|---|---|
| 920 nm 2PE | 7.2 | 11 | 0.75 | 60 | 0.66 |
| 1320 nm 3PE | 7.2 | 11 | 0.75 | 60 | 0.51 |

The cross-over depth is defined as the imaging depth where an equal amount of signal per pulse is generated using the same pulse energy at the brain surface for 1320-nm 3PE and 920-nm 2PE. Based on the ratio of 3PE to 2PE pulse energy and the EALs at 1320 nm and 920 nm, the cross-over depth can be readily calculated by solving for $z$ in **Equation 11**:

$$\frac{3\text{PE pulse energy at the focus}}{2\text{PE pulse energy at the focus}} = \frac{\text{surface 3PE pulse energy} \times \exp[-z/\text{EAL}(1320\text{nm})]}{\text{surface 2PE pulse energy} \times \exp[-z/\text{EAL}(920\text{nm})]} \tag{11}$$

For example, with EAL (1320nm) = 293 µm, and EAL (920nm) = 154 µm (**Figure 1B**), and the pulse energy ratio of 8 for GCaMP6s, the cross-over depth z is solved to be 675 µm. If the pulse energy ratio of 15 for fluorescein is used, the cross-over happens at 880 µm.

The ratio of 3PE to 2PE pulse energy can also be used to calculate the 3PE cross section based on the known 2PE cross section of the same fluorescent molecule. From **Equation 12**, the ratio of 3PE to 2PE cross section can be expressed as:

$$\frac{\sigma_3}{\sigma_2} = \frac{9}{4\pi} \tau \frac{g_p^{(2)}}{g_p^{(3)}} \frac{(\lambda_{3P})^3}{\lambda_{2P}} \frac{1}{\text{NA}^2} \left(\frac{P_{2P}}{P_{3P}}\right)^2 \left(\frac{P_{3P}}{f}\right)^{-1} \tag{12}$$

**Equation 12** shows the advantage of such a ratiometric measurement since there is no need to estimate the intracellular dye concentration $C$ or system collection efficiency $\phi$. Given the measured 2PE action cross section of GCaMP6s around 920 nm (~2×10$^{-49}$ cm$^4$s with saturated Ca$^{2+}$) (**Dana et al., 2016**), 3PE action cross section of GCaMP6s at 1320 nm is estimated to be ~3×10$^{-82}$ cm$^6$s$^2$. This result is close to the 3PE action cross section of GCaMP6f measured in an independent study (1.9×10$^{-82}$ cm$^6$s$^2$ with saturated Ca$^{2+}$) (**Macklin and Harris, 2016**).

