## [Decision Letter]

**Acceptance summary:**

Your manuscript presents a useful resource for the community about 3-photon imaging with 1320 nm excitation and 2-photon imaging with 920 nm excitation using pseudo-simultaneous imaging of 2-photon and 3-photon imaging. Careful discussions about signal-to-noise ratio in different depth and damage threshold by strong illumination will serve as an important resource for users of 3-photon microscopy.

**Decision letter after peer review:**

Thank you for submitting your article "Quantitative analysis of 1300-nm three-photon calcium imaging in the mouse brain" for consideration by *eLife*. Your article has been reviewed by two peer reviewers, one of whom is a member of our Board of Reviewing Editors, and the evaluation has been overseen by Catherine Dulac as the Senior Editor. The reviewers have opted to remain anonymous.

The reviewers have discussed the reviews with one another and the Reviewing Editor has drafted this decision to help you prepare a revised submission.

Summary:

In this manuscript, Wang et al. characterized 3-photon imaging with 1320 nm excitation and 2-photon imaging with 920 nm excitation using pseudo-simultaneous imaging of 2-photon and 3-photon imaging. They used both fluorescein in brain vasculature and neurons densely labeled with GCaMP6 for the quantification. Both signals are qualitatively same, but 3-photon shows better signal-to-noise in deep tissue (> ~500 um). They also measured tissue damages caused by heat production by the excitation. There seems to be significant heat production at ~150 mW, which could damage the brain tissue. In terms of pulse energy 1-2 nJ seems to be relatively safe for imaging in depth. Overall, this paper presents useful information about the use of 3-photon microscopy for imaging in the mouse brain.

Essential revisions:

1) As the authors note, the choice of imaging parameters (wavelength, laser repetition rate, etc.) reflects trade-offs between excitation efficiency, SBR, and tissue heating. However, these multiple parameters are only considered together very briefly in the Discussion with two examples. The utility of this work as a resource would be greatly enhanced by in-depth discussion of the interactions between different parameters wavelength, depth, FOV size, rep rate, and temperature more.

2) Interpretation is also complicated by the use of different imaging parameters in different sections of the paper. Examples include frame rate (13.6 Hz for SBR in Figure 2, vs. 2 Hz imaging for thermal analysis in Figure 3); FOV size (unstated, but apparently small in Figure 2C, vs 230 x 230 μm in Figure 3). It would be necessary to discuss how one can calibrate these differences (related to the comment #1).

3) The relationship between laser repetition rate, FOV size, and detection of Ca^2+^ transients should be clearly addressed.

---

## [Author Response]

Essential revisions:1) As the authors note, the choice of imaging parameters (wavelength, laser repetition rate, etc.) reflects trade-offs between excitation efficiency, SBR, and tissue heating. However, these multiple parameters are only considered together very briefly in the Discussion with two examples. The utility of this work as a resource would be greatly enhanced by in-depth discussion of the interactions between different parameters wavelength, depth, FOV size, rep rate, and temperature more.

We totally agree with the reviewers’ suggestion of adding a summary of the optimization for three-photon neuronal imaging with GCaMPs. In the Discussion section, we have significantly expanded the discussion on the interdependence and optimization of the imaging parameters using the results in this paper and several previous publications.

2) Interpretation is also complicated by the use of different imaging parameters in different sections of the paper. Examples include frame rate (13.6 Hz for SBR in Figure 2, vs. 2 Hz imaging for thermal analysis in Figure 3); FOV size (unstated, but apparently small in Figure 2C, vs 230 x 230 μm in Figure 3). It would be necessary to discuss how one can calibrate these differences (related to the comment #1).

In addition to the newly added discussion to address the optimization of the imaging parameters, we added the following explanation in the relevant sections to justify the choice of different imaging parameters.

In the Materials and Methods section, we have added explanations on the choice of FOV and frame rate for Figure 2 in comparing 2PM and 3PM calcium imaging:

“The calcium activities were recorded with 13.6 Hz frame rate, limited by the fastest achievable line rate of the galvanometer scanners. This frame rate is sufficiently high for imaging the temporal dynamics of GCaMP6s (τ1/e~2 s). For 2PM and 3PM comparison experiments, the FOV was reduced to fit a single neuron so that the signal-to-noise ratio is maximized to ensure accurate comparison of 2PM and 3PM calcium traces.”

In the Materials and Methods section, we have made the following modification to justify the FOV and frame rate chosen for Figure 3 for measuring the brain heating at 1320 nm:

“Anesthetized (2% isoflurane mixed with oxygen) mice were exposed to continuous scanning for 20 min at a 2-Hz frame rate and 230 μm x 230 μm FOV. The FOV was chosen to be similar to the actual achievable FOVs at the imaging depth of 1-1.2 mm (Ouzounov et al., 2017; Weisenburger et al., 2019), and the frame rate was chosen to be comparable to that in typical structural imaging. Even at 2 Hz, the frame rate is fast enough to neglect tissue cooling (~0.1 ℃/s) between successive frames (Podgorski and Ranganathan, 2016).”

In terms of scaling brain heating with FOV, we have argued in the text that “the maximum allowable imaging power can be interpolated for different imaging depths and FOVs, according to Figure 3—figure supplement 3 and Figure 3—figure supplement 4”.

In terms of scaling brain heating with frame rate, we have added the following explanation: “Brain heating is independent of the imaging frame rate when the temperature change between successive frames is negligible. For example, because the brain tissue cools at the rate of ~0.1 ℃/s (Podgorski and Ranganathan, 2016), the temperature fluctuation at a given point in the tissue due to the beam scanning is only ~0.05 ℃ at 2-Hz frame rate. Under this condition, the brain heating can be modeled as under continuous wide-field illumination.”

3) The relationship between laser repetition rate, FOV size, and detection of Ca^2+^ transients should be clearly addressed.

In the Discussion section, we first explain how the optimization of Ca^2+^ transient detection fidelity determines the maximum repetition rate at each imaging depth:

“Since three-photon imaging is essentially background-free for most of the practical imaging depths, the calcium imaging fidelity d’ depends solely on the signal strength, i.e., signal photons per neuron per second (F0). As a result, in order to maximize the number of neurons recorded, the total photon counts should be maximized within the thermal constraint and the peak intensity limit. Regardless of the imaging depth, the pulse energy at the focus should always be kept as high as possible (e.g., 1 to 2 nJ) to maximize the signal while avoiding the adverse nonlinear effects (e.g., fluorophore saturation and nonlinear damage). The maximum repetition rate is then the maximum average power allowed by the thermal constraint (e.g., as indicated in Figure 3—figure supplement 3) divided by the pulse energy at the sample surface, which can be calculated from the pulse energy at the focus using the EAL.”

Based on the maximum repetition rate, we further discuss the choice of FOV and frame rate:

“The repetition rate fundamentally limits the number of samples per second, and therefore the possible FOVs and frame rates. As a common practice, the pixel size is approximately the lateral spot size of the focus, e.g., two pixels per focal spot for a proper spatial sampling frequency. To ensure sampling uniformity when the repetition rate is low, the pixel clock is synchronized to the laser pulses, and there is an integer number of excitation pulses per pixel. Under this condition, a wide range of FOVs and frame rates combinations can be chosen as long as the repetition rate is equal to the interger multiples of the product of the number of pixels in each frame and the frame rate. In addition, the following factors should also be taken into consideration for parameter selection. Tissue heating is higher with a smaller FOV, and therefore the imaging power should be reduced accordingly with FOV (see Figure 3—figure supplement 4). For calcium imaging, the frame rate be fast enough for capturing the temporal dynamics of the calcium indicator (e.g., >5 Hz for GCaMP6s).”